# Particle-phase processing of α-pinene NO₃ secondary organic aerosol in the dark

David M. Bell[1][*], Cheng Wu[2], Amelie Bertrand[1], Emelie Graham[2], Janne Schoonbaert[1], Stamatios Giannoukos[1a], Urs Baltensperger[1], Andre S. H. Prevot[1], Ilona Riipinen[2], Imad El Haddad[1], and Claudia Mohr[2*]

[1] Paul Scherrer Institute, Laboratory of Atmospheric Chemistry, 5232 Villigen, Switzerland

[2] Department of Environmental Science, Stockholm University, Sweden

[a] Now at: ETH Zurich, Department of Chemistry and Applied Biosciences, 8093 Zurich, Switzerland

*Correspondence to*: David.Bell@psi.ch and Claudia.Mohr@aces.su.se

**Abstract.** The $NO_3$ radical represents a significant night-time oxidant present downstream of polluted environments. There are studies that investigated the formation of secondary organic aerosol (SOA) from $NO_3$ radicals focusing on yields, general composition, and hydrolysis of organonitrates. However, there is limited knowledge about how the composition of $NO_3$-derived SOA evolves as a result of particle phase reactions. Here, SOA was formed from the reaction of α-pinene with $NO_3$ radicals generated from $N_2O_5$, and the resulting SOA aged in the dark. The initial composition of $NO_3$-derived α-pinene SOA was slightly dependent upon the concentration of $N_2O_5$ injected (excess of $NO_3$ or excess of α-pinene), but was largely dominated by dimer dinitrates ($C_{20}H_{32}N_2O_{8-13}$). Oxidation reactions (e.g. $C_{20}H_{32}N_2O_8$ → $C_{20}H_{32}N_2O_9$ → $C_{20}H_{32}N_2O_{10}$ etc…) accounted for 60-70% of the particle phase reactions observed. Fragmentation reactions and dimer degradation pathways made up the remainder of the particle-phase processes occurring. The exact oxidant is not known, though suggestions are offered (e.g. $N_2O_5$, organic peroxides, or peroxy-nitrates). Hydrolysis of –ONO2 functional groups was not an important loss term during dark aging under the relative humidity conditions of our experiments (58 – 62%), and changes in the bulk organonitrate composition were likely driven by evaporation of highly nitrogenated molecules. Overall, 25-30% of the particle-phase composition changes as a function of particle-phase reactions during dark aging representing an important atmospheric aging pathway.

## 1 Introduction

Organic aerosol in the atmosphere can have important impacts on human health and climate (Jimenez et al., 2009). A substantial fraction of the organic aerosol is secondary aerosol, which is formed from reactions that lower the volatility of a molecule (Ziemann and Atkinson, 2012). One of the most prevalent families of molecules undergoing these oxidation reactions

to form secondary organic aerosol (SOA) are monoterpenes. Unlike isoprene, monoterpenes are emitted both during the day

and night (Pye et al., 2010). Because of the importance of monoterpenes at night, nocturnal oxidants ($O_3$ and $NO_3$) play an

important role in their transformation in the atmosphere (Brown and Stutz, 2012). A field study in the southeastern United

States found that ~50% of the total aerosol burden at night comes from $NO_3$ radical derived chemistry (Xu et al., 2015;Xu et

al., 2014;Ayres et al., 2015;Lee et al., 2016). Other studies have shown the importance of nitrate chemistry in both urban and

pristine areas with monoterpenes (Yan et al., 2016;Stefenelli et al., 2019;Huang et al., 2019;Kiendler-Scharr et al., 2016).

Although α-pinene is globally the most abundant monoterpene (Guenther et al., 2000), β-pinene has been the most

studied of monoterpenes interacting with $NO_3$ radicals because of resulting high SOA yields (Takeuchi and Ng, 2019;Nah et

al., 2016;Boyd et al., 2015) relative to α-pinene. β-pinene was estimated to be the precursor of up to ~20% of the total nighttime

OA in the Southeastern US (Ayres et al., 2015). However, recent studies show that SOA formation from α-pinene + $NO_3$ is

also possible (Romonosky et al., 2017;Takeuchi and Ng, 2019;Nah et al., 2016). In chamber studies, SOA from both α-pinene

and β-pinene + $NO_3$ contains a substantial fraction of dimers when the radical balance is dominated by $RO_2$ + $NO_3$ or $RO_2$ +

$RO_2$ reactions (Takeuchi and Ng, 2019;Claflin and Ziemann, 2018), while monomers are the dominant products when $RO_2$ +

$HO_2$ is the dominant pathway (Nah et al., 2016). When experiments were conducted in the dark, monoterpene + $NO_3$ products

were shown to steadily evolve, with a steady change in O:C and N:C ratios reported by Nah et al. (2016). In Takeuchi and Ng

(2019) ~9-17% of organo-nitrates from either α-pinene or β-pinene SOA hydrolyze at moderate relative humidities (~ 50%

RH). The studies of Nah et al. (2016) relied on acidic seeds, which could impact the particle phase aging (Riva et al., 2019),

when compared to neutral seeds or homogeneously nucleated SOA. Alternatively, Claflin and Ziemann (2018) speculated

about the importance of dimer formation within the particle-phase itself for β-pinene + $NO_3$ SOA. However, quantifying the

absolute magnitude of the importance of the particle-phase processes was not possible in any of these studies. These contrasting

studies demonstrate the processes occurring during dark aging of $NO_3$ derived SOA are still not well understood, and warrants

further studies to improve our knowledge on the processes occurring during dark aging of $NO_3$ derived SOA.

The ability to observe the chemical changes in real time during aging have been limited to instruments with significant

fragmentation, low time resolution, or limited to off-line analysis. The advent of new soft-ionization techniques makes it

possible to follow these changes in real-time with high molecular resolution (Pospisilova et al., 2020). We will employ the

extractive electrospray ionization time-of-flight mass spectrometer (EESI-TOF) along with a chemical ionization mass spectrometer with filter inlet for gases and aerosols (FIGAERO-CIMS) to elucidate the particle-phase composition with high chemical and temporal resolution and uncover the changes occurring therein. Here, we investigate the composition of SOA formed from α-pinene + $NO_3$ in dark conditions, determine the processes occurring in the particle phase, separate them from products formed in the gas phase, and determine the magnitude of the effect dark aging has on the composition of $NO_3$ derived α-pinene SOA.

## 2. Experimental

### 2.1 Chamber Conditions

A series of batch-mode chamber experiments (Table 1) investigating formation and aging of SOA formed from $NO_3$-initiated chemistry was performed in the 8 m³ Teflon chamber at the Paul Scherrer Institute, Switzerland, described previously (Platt et al., 2013). Measurements were performed with a proton-transfer-reaction with a quadrupole mass spectrometer (PTR-MS, Ionicon), extractive electrospray inlet coupled to a long time-of-flight mass spectrometer (EESI-ToF, Tofwerk), a scanning mobility particle sizer (SMPS, TSI model 3938), a chemical ionization mass spectrometer with filter inlet for gases and aerosols (FIGAERO-CIMS, Aerodyne), a thermo-denuder coupled with an SMPS, an ozone gas monitor (Thermo 49C), and a $NO_x$ monitor (Thermo 42C). A zero air generator (AADCO) was used to supply clean air to the chamber and instrumentation. Isothermal evaporation chambers were used for the investigation of particle volatility. The FIGAERO-CIMS sampled on its own dedicated line from the atmospheric simulation chamber with a flow rate of 5 L min⁻¹. Gas monitors and PTR-MS operated behind a heated (80 °C) 3 m stainless steel sampling line with a 5 s residence time. The particle instruments sampled from a 3 m stainless steel line with ~3 s residence time. The chamber was cleaned after every experiment and overnight by purging it with zero air (50 L min⁻¹ input and 50 L min⁻¹ output), and heating it to 30 °C. In the morning the chamber was expanded to its full volume with zero air. The chamber was deemed clean if the particulate concentrations overnight were less than 50 cm⁻³ and $NO_x$ levels below 30 ppb. A persistent contamination of cresol (~1 ppb) was present in the chamber from experiments that took place prior to those with α-pinene.

Five experiments generated α-pinene SOA from reactions with $NO_3$ radicals at 58 ± 5% relative humidity (RH) and 21 ± 3 °C. No seed particles were used in this study, in contrast to other studies that have been performed on similar systems

(Takeuchi and Ng, 2019). $NO_3$ radicals were formed from the decomposition of $N_2O_5$, which was injected by passing dry clean

air over a solid sample of $N_2O_5$. For blank experiments, α-pinene was injected ~30 min. after $N_2O_5$. In these experiments, no

new particles were formed after the $N_2O_5$ injection and prior to α-pinene injection; also the gaseous cresol contamination did

not lead to particle formation during blank experiments. However, in each experiment, the contamination was observed by

both the FIGAERO-CIMS and EESI-ToF after enough organic mass had formed to allow the contaminants to partition into

the particle phase; the cresol contamination constitute ~1-2% of the total EESI-ToF signal and ~5% of the total FIGAERO

signal. In all experiments, new particle formation occurred promptly after injection of $N_2O_5$ when a VOC was present in the

chamber. α-pinene was injected volumetrically with the PTR-MS following the absolute concentration; in experiments 1 – 3,

~100 ppb of α-pinene were injected, and 20 ppb of α-pinene in Experiments 4 – 5. It was intended that Experiments 1 and 3

would be identical repeats since the α-pinene and $N_2O_5$ additions were effectively identical, however there was large

differences in the mass loading observed. The reason for the difference in yield is not clear, and may result from

inhomogeneities in the chamber during the short burst of $N_2O_5$. Therefore, even though experiments 1 and 3 were intended to

be conducted under similar conditions, we cannot state unequivocally that they are identical.  The FIGAERO-CIMS and EESI-

ToF were only present for experiments 1-3, which will be the focus of the discussion below. Even though the FIGAERO-

CIMS and EESI-ToF were not present for experiments 4 and 5, these experiments are included to confirm the changes in

particle mass concentration with dark aging at lower mass concentrations are consistent with the measurements at higher mass

concentration.

The Framework for 0-D Atmospheric Modeling (F0AM) (Wolfe et al., 2016) with the Master Chemical Mechanism

(MCM) (Saunders et al., 2003;Jenkin et al., 1997) was used to model the gas-phase and radical chemistry in the chamber.

F0AM was used to estimate the amount of $N_2O_5$ injected into the chamber, and in all cases the concentration of $N_2O_5$ was

between 80 and 300 ppb. During dark aging (3-4 h) the volume of the chamber was reduced by instruments sampling from the

chamber, and no additional air was added to the chamber during this time.

**Table 1)** Experimental parameters for all experiments. [a]Measured by the SMPS, [b]modeled $N_2O_5$ concentration based

on VOC decay assuming $1.2 \times 10^{-12}$ $sec^{-1}$ as the $NO_3$ + α-pinene rate constant, [c]volumetric addition of α-pinene.

| Experiment # | Maximum SOA[a] ($\mu g$ $m^{-3}$) | $N_2O_5$[b] (ppb) | RH (%) | Temp | α-pinene injected[c] (ppb) | Particle Mass Spectrometers Operational |
|---|---|---|---|---|---|---|
| 1 | 19 | 300 | 58 | 21 | 100 | EESI (semi-operational) and FIGAERO-CIMS |
| 2 | 39 | 80 | 60 | 22 | 100 | EESI and FIGAERO-CIMS (first filter performed later) |
| 3 | 62 | 300 | 58 | 18 | 100 | EESI and FIGAERO-CIMS |
| 4 | 8 | 400 | 58 | 20 | 20 | -none- |
| 5 | 7 | 200 | 62 | 20 | 20 | -none- |

In all experiments, the mass measured by the SMPS was corrected for particle wall loss. The wall loss rate ($k_{wall}$) was

calculated from the exponential decay of the total particle number concentration ($cm^{-3}$) measured by the SMPS, corrected for

coagulation. The particulate wall loss is defined by:

$$\frac{dN}{dt} = -k_{coag}N^2 - k_{wall}N \qquad \text{Eq 1.}$$

where $N$ is the particle number concentration and $k_{coag}$ corresponds to the coagulation coefficient ($5E-10$ $s^{-1}$). The number of

particles lost during the measurement time was scaled according to the mean mass of the entire particle population, which is

based on the geometric mean mobility diameter ($d_m$) from the measurement time, assuming a density of 1.2 g $cm^{-3}$. The wall

loss-corrected mass was divided by the uncorrected mass to obtain the wall loss correction factor that was applied to the EESI-

ToF and FIGAERO-CIMS data to correct for particle wall loss, during experiments 1-3.

## 2.2 EESI-ToF – Extractive Electrospray Ionization Time-of-Flight Mass Spectrometer

The EESI-ToF measured the molecular constituents of the SOA formed in the chamber, sampling the aerosols at 1 L min$^{-1}$. The aerosol sample passed through a multi-channel denuder that strips the gas phase species from the particles, with minimal evaporation of the particles (Lopez-Hilfiker et al., 2019). The aerosol sample intersected with a spray of droplets (50:50, $H_2O$ : acetonitrile) doped with 100 ppm NaI which emanated from an electrospray probe. The soluble portion of the aerosol was extracted into the liquid, and after the droplet evaporated, the molecules were observed in the mass spectrometer as $Na^+$-adducts. The adducts were guided through a series of ion-guides and are separated based on their mass-to-charge ratio in a time-of-flight mass spectrometer. In the experiments presented here, the resolution of the mass spectrometer was $5,500 – 7,000$. Background measurements were continually and repeatedly performed by sampling the contents of the atmospheric simulation chamber via a particulate filter (e.g. 4 min. chamber air without and 1 min. with filter). A noise filter was used to remove data points impacted by severe electronic noise, which was periodically observed throughout the campaign. The reported signal was a result of subtracting the background filter periods from the adjacent sampling periods. A filtering threshold was applied to the EESI-ToF data, where signals that were $1\sigma$ (standard deviation) greater than background data were considered statistically significant and included in the analysis. The EESI-ToF time series were averaged to 5 min resolution, including one sample and filter period cycle. The EESI-ToF signal was scaled according to the molecular weight of each respective molecule $i$ and converted to a mass flow rate using:

$$EESI\ (ag\ s^{-1}) = \sum_i \frac{EESI(Hz)_i \cdot MW_i \cdot 10^{18}}{6.023E23}$$

where EESI(Hz) is the raw data from the TOF, and $MW_i$ is the molecular weight, normalized using the Avogadro's number, and a conversion factor from grams to attograms. Within the EESI-ToF, the detection efficiency varied depending on a number of factors including particle size (Lee et al., 2021), extraction efficiency in the EESI-ToF droplet, and mass transmission of an ion from the inlet to the mass spectrometer. Despite all these potential impacts, if all ions were measured equally, the overall agreement between measured EESI-ToF signal (attograms per second - ag s$^{-1}$) followed a linear trend with measured organic mass (SMPS) despite day-to-day variations in absolute intensity (Figure S2). Experiments 2 and 3 fell along the same slope, because the EESI-ToF capillary and TOF settings were not adjusted between experiments. In contrast, experiment 1 had a much lower slope because the EESI-ToF was still being optimized (capillary position and TOF settings were being altered)

and had low sensitivity during this experiment. Consequently, bulk information was obtained by long averaging times for experiment 1, but time series of individual molecular formulae was not used due to the noisy signal.

A majority of the ion signal (>95%) was from $Na^+$-adducts. However, a consequence of using acetonitrile ($C_2H_3N$) as a solvent, there was also a small fraction of the signal that is associated with $Na^+$-adducts+acetonitrile. For instance, $Na^+$-$C_{20}H_{32}N_2O_8$ was one of the main ions observed for α-pinene SOA, and there was an associated peak for $Na^+$-$C_{22}H_{35}N_3O_8$

($C_{20}H_{32}N_2O_8 + C_2H_3N$) that results from the interaction between spray solution and the constituent of SOA. For all $C_{20}H_{32}N_2O_x$ molecules, the acetonitrile cluster represented 1-3% of the parent ion. For the cases where unambiguous clustering occured (e.g. $C_{20} + C_2H_3N$), $C_2H_3N$ was subtracted from the molecular formula. For some compounds, the molecular formulae of the clusters overlapped with $C_{20}$ oxidation products (e.g. $C_{18}H_{28-32}N_{0-2}O_x$). As the $C_{18}$ molecules represented less than 1% of the total signal, the possible contribution of the acetonitrile cluster was not corrected for. Because there was not a large variety in

the carbon distribution of the SOA formed, as shown in the results below, a majority of the mass was corrected for acetonitrile clustering.

One of the most prevalent peaks in the mass spectra of the EESI-ToF was the $C_{10}H_{16}O_2$ (~50-65% total signal), which resulted from an artefact of specific dimer degradation in the electrospray droplet. A follow-up paper will describe details the corrections for the artefact and our confidence in identifying it. In brief, the time series of the $C_{10}H_{16}O_2$ molecule was strongly

correlated ($R^2 > 0.96$) only with $C_{20}H_{32}N_2O_8$, shown in Fig. S1 (inset), even during evaporation experiments. Based on this correlation, $C_{20}H_{32}N_2O_8$ likely fragments in the electrospray droplet to form two $C_{10}H_{16}O_2$ species, which were observed, while the –$ONO_2$ groups were lost and not detected. If we presume two $C_{10}H_{16}O_2$ molecules come from a single $C_{20}H_{32}N_2O_8$ and correct the mass loss for the nitrate groups lost, then there is good agreement with the measured mass. Therefore, the total contribution of $C_{10}H_{16}O_2$ was converted to $C_{20}H_{32}N_2O_8$. Additional potential measurement artefacts included loss of $HNO_3$

(Liu et al., 2019), though, this was a minor pathway making up <1% of the parent ion signal for any molecular formula (e.g. $C_{20}H_{32}N_2O_x \rightarrow C_{20}H_{31}NO_{x-3} + HNO_3$).

### 2.3 FIGAERO-CIMS - Chemical ionization mass spectrometer with filter inlet for gases and aerosols

The molecular composition of organic compounds in gas and particle phases was measured using a FIGAERO-CIMS. The FIGAERO inlet was coupled with a high-resolution time-of-flight chemical-ionization mass spectrometer (HR-ToF-

CIMS) using I- as reagent ion and an X-ray generator as ion source. The resolution of the mass spectrometer was 5,000 –

6,000. The design and operation of the FIGAERO-CIMS were similar to that described in previous studies (Huang et al.,

2019;Lopez-Hilfiker et al., 2014). Briefly, particles were collected on a 25mm Zefluor® PTFE filter (Pall Corp.) via a sampling

port (flow rate 5 L min$^{-1}$). The duration of particle-phase sampling depended on the mass loadings and was 10 – 20 min for

most of the experiments. During the particle-phase sampling, gases were directly measured by the CIMS from the chamber

via a Teflon line at 5 L min$^{-1}$. When the particle phase sampling was done, the gas-phase measurement was switched off and

particles on the filter were desorbed by a flow of gradually heated ultra-high-purity (99.999 %) nitrogen and transported into

the CIMS. A FIGAERO-CIMS desorption round lasted about 40 min: 20 min of ramping temperature of the nitrogen flow up

to 200 $^{\circ}$C, followed by a 20 min "soak period" at a constant 200 $^{\circ}$C. After that it was cooled down to room temperature. The

resulting mass spectral signal evolutions during a desorption round as a function of desorption temperature are called

thermograms (Lopez-Hilfiker et al., 2014). The raw FIGAERO-CIMS data were analysed with Tofware (Aerodyne Research,

Inc. and Tofwerk AG) and codes written with MATLAB. The integration of thermograms of individual compounds yields

their total signal in counts per deposition. For the first filter in Exp. 2, due to a software failure, the filter was stored wrapped

in aluminum foil for ~7 h before the being desorbed. However, comparing the first filter with others, the changes in O/C, N/C

as well as the ratio of monomers/dimers were similar to other experiments.  For further details about the data analysis, see Wu

et al. (2021).

## 3 Results

### 3.1 Initial particle molecular composition and instrumental comparison

In the experiments performed here, the α-pinene rapidly reacts with $NO_3$, with corresponding prompt formation of

organic aerosol mass measured by the SMPS (shown in Figure 1A). After α-pinene consumption and particle formation, the

SOA mass steadily decays by evaporation (Figure 1B), which is consistent with previous studies (Nah et al., 2016). In all

experiments, except experiment 2, the α-pinene is fully consumed, while in experiment 2 there are ~20 ppb of unreacted α-

pinene left over in the chamber after the $N_2O_5$ is fully consumed, because the injection of $N_2O_5$ was less than the other

experiments. The incomplete consumption of α-pinene comes from a injection of $N_2O_5$ that was less than the total concentration

of α-pinene. The relatively small concentrations of $N_2O_5$ injected also changes the radical balance in the chamber, which will

be discussed in the next section. In the other experiments with an excess of $N_2O_5$ injected, hydrolysis on the walls likely represents a significant sink for any unreacted $N_2O_5$. During the experiments presented here, the analysis will focus on the higher loading experiments (experiments 1-3) because both the FIGAERO-CIMS and EESI-ToF were not present for experiments 4 and 5 (see Table 1).

In this section we compare the chemical composition measured by both the EESI-ToF and FIGAERO-CIMS. Detailed 200 mass spectra averaged from experiments 1 – 3 for both the EESI-ToF and FIGAERO-CIMS are presented in Figures 2A and B over the first 10-46 min after particle formation. Figure 2A also colors the composition according to the number of carbon atoms in the molecular formulae. Dinitrates dominate the dimer fraction of α-pinene SOA, with molecular formulae of $C_{20}H_{32}N_2O_{8-13}$ making up ~60 – 85% of the total composition for the EESI-ToF and 45 – 56% for the FIGAERO-CIMS (the range of values comes from the variation across experiments 1 – 3 for the beginning of each experiment), consistent with 205 previous observations (Takeuchi and Ng, 2019), and with dominant $RO_2$ + $RO_2$ reactions shown for other systems (Zhao et al., 2018;Molteni et al., 2019). There are slight differences when comparing the carbon number distribution observed by both the FIGAERO-CIMS and the EESI-ToF, shown in Figures 2C and D, reflected in the EESI-ToF detecting more dimers over monomers, when compared to the FIGAERO-CIMS. Additionally, singly nitrated monomers ($C_{10}H_{15}NO_{5-10}$) are observed by both instruments (5-7% EESI-ToF and 7-10% FIGAERO-CIMS), while monomer dinitrates ($C_{10}H_{14,16}N_2O_{7-11}$) are mostly only 210 observed by the FIGAERO-CIMS and are slightly above background levels in the EESI-ToF for experiment 3. The monomer dinitrates have the highest concentrations in experiments 1 (10%) and 3 (12%), while they are substantially lower in experiment 2 (2%). Monomer dinitrates likely form via an $RO_2$ + $NO_2$ reaction to form peroxynitrate functional groups(Chan et al., 2010), which is speculated to form monomer trinitrates in the isoprene + $NO_3$ system (Zhao et al., 2020). The lack of their formation in experiment 2 agrees with the smaller concentrations of $N_2O_5$ injected, and corresponding smaller amount of $NO_2$ present.

The main differences between the instruments come from the oxygen distribution of the dinitrate dimers ($C_{20}H_{32}N_2O_{8-13}$) where the most prevalent molecular formulae are with #O = 8 for the EESI-ToF and #O = 9 for the FIGAERO-CIMS, shown in Figures 3A and 3B. This is similar to differences observed between offline ESI and online FIGAERO-CIMS measurements for β-pinene SOA (Takeuchi and Ng, 2019;Claflin and Ziemann, 2018). As mentioned above, there is a small contamination from cresol ($C_7H_8O$) in the chamber, which shows itself in a series of $C_7$ and $C_{17}$ molecules present in the mass

spectra (in experiment 2: 6% in the FIGAERO-CIMS and 1% in the EESI-ToF). The contaminant molecule with the largest signal in both instruments is $C_{17}H_{26}N_2O_{11}$, from a dimer formed between a cresol monomer and an $\alpha$-pinene monomer. The changes of the contaminants with dark aging time is relatively minor with respect to all other particle-phase components and therefore they are spectators during aging.

Overall, the composition generally agrees between both instruments very well, demonstrating relatively good overlap in the main molecular classes observed (dimer dinitrates and singly nitrated monomers) making up ~65-90% of the total SOA composition in both instruments. Observed differences do not affect the general scope of the paper focused on the intra-particle reactions of organonitrate species, as will be shown in the following sections. Additionally, the similarity in the measured composition in experiments 1 and 3 suggests these conditions were relatively similar, though they should not be considered exact replicates because of the difference in the SOA yield between the experiments (Table 1).

## 3.2 Gas-Phase Radical Chemistry and Impact on Particle Composition

Because particle-phase reactions are not well understood, it is necessary to understand what radical pathways controls the initial composition of the SOA. The radical chemistry of $NO_3$ is important to consider, because Ng et al. (2008) demonstrated that the balance of $NO_3$ and $RO_2$ radical chemistry plays an important role in the yields of formation of SOA from isoprene + $NO_3$. In their study, larger SOA yields were observed under an $RO_2$ + $RO_2$ dominant regime compared to an $RO_2$ + $NO_3$ dominant regime. The $RO_2$ chemistry regime promotes the formation of dimers with lower volatility increasing yields of SOA, and is consistent with the large prevalence of ROOR dimers observed in isoprene + $NO_3$ SOA (Ng et al., 2008). For monoterpenes, only minor differences were found in the yields of $\beta$-pinene + $NO_3$ between the $RO_2$ + $NO_3$ vs. $RO_2$ + $HO_2$ regimes, however, the $RO_2$ + $RO_2$ dominant regime was not investigated (Boyd et al., 2015).

F0AM was used here to model the fate of $RO_2$ radicals assuming generalized rate constants for the known $RO_2$ radicals formed via the reaction of $\alpha$-pinene + $NO_3$. The MCM does not include intra-molecular hydrogen shifts, auto-oxidation reactions, nor ROOR formation, but despite these limitations, the MCM provides general insight into the predicted radical chemistry regimes. Based on the modelling, shown in Figure S3, $RO_2$ + $NO_3$ reactions are predicted to be the dominant pathway in the excess of $N_2O_5$ (experiments 1 and 3), while in the excess of $\alpha$-pinene, $RO_2$ + $RO_2$ reactions will dominate (experiment

2). HO$_2$ is not an important reaction partner since there are limited formation mechanisms under our conditions (Boyd et al.,

2015).

The first generation RO$_2$ radical from the reaction of α-pinene + NO$_3$ is C$_{10}$H$_{16}$NO$_5$, which can react with itself via

the ROOR pathway and form C$_{20}$H$_{32}$N$_2$O$_8$ + O$_2$. Based on this reaction pathway the C$_{20}$H$_{32}$N$_2$O$_8$ likely contains two –ONO$_2$

functional groups and a peroxy linkage.  This molecule is also the dominant dimer that is observed in the gas-phase at the

beginning of the experiment, shown in Figure 3C, and observed by both instruments in the particle-phase.  More oxygenated

dimers (C$_{20}$H$_{32}$N$_2$O$_{9,10}$) are also observed in the gas-phase (Figure 3C), presumably arising from a similar dimer formation

mechanism (RO$_2$ + RO$_2$), but with more highly oxygenated RO$_2$ radicals that can form from the initial RO$_2$ through

autooxidation (Bianchi et al., 2019) or the alkoxy pathway (Molteni et al., 2019). Highly oxygenated dinitrate C$_{20}$ molecules

detected here have also been observed in the gas-phase in pristine environments (Yan et al., 2016). Gas-phase measurements

of C$_{20}$H$_{32}$N$_2$O$_{8-10}$ in Figure 3C importantly show the presence of dimers in the gas phase, demonstrating they are not exclusively

formed in the particle phase, as hypothesized for β-pinene + NO$_3$ (Claflin and Ziemann, 2018).

Since we are operating in different predicted radical regimes, RO$_2$ + NO$_3$ dominated (experiments 1 & 3), and RO$_2$ +

RO$_2$ dominated (experiment 2), there should be differences in the molecules that are observed in the respective experiments.

In experiment 2 (RO$_2$ + RO$_2$ dominant), there is a larger fraction of dimers measured by both instruments (Figures 2C and

2D), in comparison to the other experiments (1 and 3) where RO$_2$ + NO$_3$ reactions are predicted to be the dominant pathway.

Consistent with this observation, when dimer formation is suppressed (experiments 1 and 3) the monomer fraction has a

contribution of 17-30% compared to 5-7% in experiment 2. Given the dominant fraction of dimers in all experiments, the

importance of RO$_2$ +RO$_2$ reactions is underestimated, and the RO$_2$ + NO$_3$ reactivity is overestimated by the MCM, especially

in experiments 1 and 3 (see Fig. S3). Despite the potential differences in the predicted radical regime, formation of dimers

dominates the particle-phase composition, meaning RO$_2$ + RO$_2$ reactions are always very important even in experiments where

RO$_2$ + NO$_3$ is predicted to be the dominant pathway. These findings are consistent with the importance of dimers formed from

RO$_2$ + RO$_2$ reactions in other systems (Berndt et al., 2018a;Berndt et al., 2018b;Zhao et al., 2018;Ng et al., 2008;Molteni et

al., 2019;Rissanen et al., 2015;Simon et al., 2020;Heinritzi et al., 2020), and with higher SOA yields from NO$_3$ initiated

oxidation during dominant RO$_2$ + RO$_2$ chemistry (Ng et al., 2008;Bates et al., 2021).

### 3.3 Evolution in the Particle Phase Composition

Figure 1B shows that after formation of SOA the particles steadily evaporate in the chamber, suggesting subsequent changes in composition occurs via particle evaporation and/or other particle phase processes. The high time-resolution and detailed molecular information determined by the EESI-ToF, and the FIGAERO-CIMS, were used to determine the processes occurring and their time scales.

A comparison of the temporal evolution of the elemental N:C vs. O:C ratios for experiments 1-3 from EESI-ToF data
is illustrated in Figure 4A, colored according to experimental time. Results from experiment 1 are noisy due to the low sensitivity (see Figure S2), but the data is included because of the similar trend with the other experiments. The FIGAERO-CIMS is not included because of the low time resolution, which would include only 2 or 3 data points. In Figure 4A, there is a trend toward increasing O:C during the experiment, coupled with a decreasing N:C that correlates with mass concentration.

Considering each nitrogen atom is likely part of an $-ONO_2$ functional group (Takeuchi and Ng, 2019), the decrease
in N:C suggests the increase in oxidation state (O:C) as a function of aging is larger than the apparent change shown in Figure 4A. Accounting for the substantial fraction of oxygen in the $-ONO_2$ functional groups, the O:C ratio for the non-nitrate functional group portion is determined by: $\#O_{No-Nitrogen} = \#O_{measured} - 3*\#N_{measured}$. With the contribution of the nitrate functional groups removed the O:C ratio (shown in Figure 4B) increases by 25-60% during the course of dark aging, demonstrating the components of SOA either undergo further oxidation during dark aging, or the particles´ O:C increases as a function of
evaporation of compounds with lower O:C ratios from the particles.

If evaporation was the sole reason for changes in the oxidation state of the SOA, then there should be limited changes in the chemical composition as a function of aging except for those associated with evaporation (i.e. there should be nothing forming in the particle phase). Dimer dinitrates ($C_{20}H_{32}N_2O_{8-13}$) make up between 45 - 85% of the total ion signal through the whole dark aging time period, and will be the focus of the initial discussion on particle-phase processing. The wall-loss
corrected time series of all $C_{20}H_{32}N_2O_x$ molecules over the first 180 min of the experiment is shown in Figure 5A for experiment 2 (with a similar plot for experment 3 shown in Figure S4, experiment 1 is not included because of the high relative noise). All molecules exhibit prompt incorporation in SOA, followed by steady changes, with molecules both increasing and decreasing in intensity with time. The #O = 8 molecule steadily decays after particle formation with a decay rate of 0.01 min$^{-1}$, while the

#O = 9 steadily increases by 80% over 3 hours. To illustrate the changes more clearly, Figure 5B shows the changes in the composition relative to t = 10 min after particle formation, when the maximum in particle mass is reached. The #O = 10 – 11 dinitrates increase 150 – 450% during 3 h of dark aging, the contribution of these molecular formulae is ~1% at the beginning of the experiment and increases to ~2-5% of the total EESI-ToF signal. The FIGAERO-CIMS observes qualitatively similar changes in the composition as a function of dark aging with the $C_{20}H_{32}N_2O_{8-13}$ molecules, though the magnitude of the changes is smaller than those shown in Figure 5A and B. Overall, Figure 5A and B demonstrates production of more highly oxygenated dinitrate dimers in the particle phase. The production observed in experiment 2 is also observed in experiment 3, meaning it occurs regardless of the presence of excess $NO_3$. Therefore, the particle-phase production is not due to gas-phase oxidation reactions with excess $NO_3$ and subsequent condensation to the particle phase, but it must be occurring in the particle-phase.

Since we ruled out gas-phase reactions, production of higher oxygenated dimers could result either from monomer conversion to dimers, or further oxidation in the particle phase. Figures 6A and 6B compare the carbon distribution of both the EESI-ToF and FIGAERO-CIMS and how they change as a function of dark aging for experiment 3. This experiment is used because there is the best overlap between the instruments (experiment 1 is also provided in Figure S5). In both the EESI-ToF and FIGAERO-CIMS, the main species decaying in the particle phase are $C_{20}$ dimers with additional contribution from the decays of $C_{10}$ molecules. Formation is dominated by other $C_{20}$ molecules and some $C_{10}$ molecules, with both instruments showing the formation of more highly oxygenated $C_{20}$ molecules (Figure 6C and 6D). The steady evaporation and lack of oxidant in the gas phase likely precludes heterogeneous dimer formation from further gas-particle conversion. We cannot rule out the possibility that dimers decay to monomers, and then undergo dimerization reactions to reform as higher oxygenated dimers. Though, if the pathway to more highly oxygenated dimers occurred via degradation of dimers to monomers with subsequent dimer formation, then there would be a shift to less oxygenated monomers to compensate this effect. The oxygen distribution of the $C_{10}$ molecules for the EESI-ToF is shown as a function of formation and depletion in the supplement and they do not present a coherent change with dark aging (Figure S6).

The argument from the preceding paragraph assumes that the sensitivity of the EESI-ToF holds equally for all molecules. On a molecule-by-molecule basis there is roughly a spread of 1 order of magnitude in the sensitivity of the EESI-ToF toward different ions (Wang et al., 2021;Lopez-Hilfiker et al., 2019), but when comparing bulk SOA composition there

is good agreement with measured mass here (Fig. S2) and during measurements performed in Zurich, Switzerland (Qi et al.,

2019;Stefenelli et al., 2019). Additionally, the SOA composition measured by the EESI-ToF compares well to the predicted

particle composition based on gas-particle partitioning, using gas-phase measurements performed by the PTR3 (Surdu et al.,

2021). This provides confidence that the EESI-ToF is not missing specific molecules here. However, even if the EESI-ToF

were not sensitive to the species that are forming the more highly oxygenated molecules (i.e. monomer exchange reactions),

then the formation of higher oxygenated dimers ($C_{20}H_{32}N_2O_{9-11}$) would have to be nearly equally balanced by the consumption

/ disappearance of $C_{20}H_{32}N_2O_8$ (in order for the SMPS mass and EESI to agree in Fig. S2). Further, the FIGAERO-CIMS

would have to be biased in the exact same way as the EESI-ToF. Considering the agreement presented here and in previous

studies it is more likely that the combination of the EESI and FIGAERO are capturing the change in chemical composition.

Therefore, the species initially formed in the gas-phase appear to undergo particle-phase oxidation to form more

highly oxygenated molecules observed in Figures 5 and 6. The other important particle-phase reaction pathways shown in

Figure 6 include fragmentation reactions (e.g. $C_{20} \rightarrow C_{19}$) and dimer degradation to monomers, which will be discussed further

in terms of their absolute magnitude in the follow section.

### 3.4 Absolute Magnitude of Particle-Phase Reactions

Using Figures 5 and 6 it is possible to assess the magnitude of the processes taking place in the particle phase by

taking all molecules observed by the EESI-ToF that are formed relative to $t = 10$ min and creating a mass balance during the

whole dark aging period. Figure 7 shows the absolute intensity of all molecules detected by the EESI-ToF from experiment 2,

highlighting the fraction of molecules increasing after the maximum in the organic mass concentration (shown in light blue),

the signal of those molecular formulae that decrease (in orange, of which $C_{20}H_{32}N_2O_8$ makes up the great majority), and the

initial contribution of the molecules that increase (dark blue). The molecules formed during the dark aging period make up

~30% of the total aerosol composition after 3 hours, representing a significant shift in the composition of SOA from the initially

formed species. The formation found in the particle phase for experiment 3 is ~25% (shown in Figure S7), and is similar to

that reported in Figure 7. If there was formation of volatile species, then this would not be captured by the current treatment

and the magnitude of particle-phase processes would be even larger. Therefore, the reported 25-30% formation via particle-

phase processes would be closer to 50% if evaporation was resulting from particle-phase reactions and not solely from volatile

components repartitioning. Additionally, this treatment also does not account for potential step-wise oxidation reactions (e.g. $C_{20}H_{32}N_2O_8 \rightarrow C_{20}H_{32}N_2O_9 \rightarrow C_{20}H_{32}N_2O_{10}$), which would again result in an underestimate of the importance of particle-phase reactions. Therefore, a lower estimate of 25 – 30% of the total composition of α-pinene SOA is altered as a function of particle-phase processes over the 3-hour experimental time.

Figures 7 and S7 also show that particle-phase processes appear to be the most important over the first 2-2.5 hours of dark aging. Nearly all of the changes in the particle composition observed by the EESI-ToF come at the expense of $C_{20}H_{32}N_2O_8$, which diminishes from 70% of the total composition to ~35% after 3 hours of aging. Assuming the steady formation of higher oxidation products is the result of a reaction with $C_{20}H_{32}N_2O_8$, the summation of signal forming after $t$ = 10 min in Figure 5A represents the fraction of $C_{20}H_{32}N_2O_8$ consumed via particle-phase processes. In experiment 2, the $C_{20}H_{32}N_2O_8$ has a mass flux of 1.05 ag s$^{-1}$ and it decreases to 0.61 ag s$^{-1}$ after 3 hours, with a corresponding increase of oxidation products from 0.17 ag s$^{-1}$ to 0.38 ag s$^{-1}$ over that same time frame (shown in Figure 5A). Approximately half of the total depletion observed arises from an increase in oxidation, with the remainder coming from evaporative losses.

Although oxidation reactions account for a large majority of the particle-phase processes occurring, Figure 6A-D shows it is not the only process occurring. The increases in the oxidation state of the $C_{20}$ dimers make up 60-70% (range for experiments 2 and 3) of the total fraction of the species formed in the particle phase. Using analysis similar to the results shown in Figure 6A-D, we can separate the other processes according to how the carbon distribution changes. The other two minor pathways observed by the EESI-ToF are: increases in the monomer fraction (10-15% to the total changes), and formation of $C_{8,9}$ or $C_{18,19}$ via fragmentation reactions from $C_{10}$ or $C_{20}$ molecules (10-20%). Fragmentation reactions will also lead to the formation of smaller molecules containing 1-2 carbon atoms, which should rapidly evaporate from the particle phase. Assuming the molecules leaving the particle phase have a molecular formula of $CH_2O$, as a simple example, it would represent ~10% loss of mass relative to $C_{20}H_{32}N_2O_8$. The amount of mass evaporating from the particle from fragmentation reactions would be on the order of 0.2 – 0.3% of the total mass (or 0.1 μg m$^{-3}$ in the experiment shown in Figure 8). Therefore, fragmentation reactions are not responsible for a significant loss of mass during dark aging. We should note, we do observe a few molecules increasing more than 25% in the gas-phase consistent with fragmentation reactions, including: $C_2H_4O_3$, $C_3H_6O_3$, $C_4H_8O_3$, $C_3H_5NO_5$, $C_4H_7NO_5$, $C_5H_9NO_5$.

In Figure 4A we show the N:C ratio decreases during dark aging. Removal of dimer dinitrates results in a significant mass change during the dark aging period (10 – 30% of the total signal). Due to the significant nitrogen content of these dinitrate dimers, they will significantly impact the N:C ratio during aging. Figure 1b shows that the SOA evaporates during aging in the chamber. Evaporative loss of dinitrate dimers is possible, though not common considering dimers typically have very low volatility. However, when performing isothermal evaporation measurements in the atmospheric simulation chamber both the $C_{20}H_{32}N_2O_{8,9}$ are susceptible to repartitioning (Figure S8). Because these dimers are low-volatility to semi-volatile molecules they are in equilibrium between the particles, gas-phase, and the walls (Bertrand et al., 2018;Krechmer et al., 2020). Therefore, part of the loss of $C_{20}H_{32}N_2O_{8,9}$ is due to repartitioning and accounts for a significant fraction of the organonitrates lost from the particle phase observed by the EESI-ToF, and a significant fraction of the decreasing signal in the FIGAERO-CIMS. Previously, the loss of $NO_3$ from the particle phase has been assumed to come from hydrolysis of $-ONO_2$ functional groups due to the loss of $NO_3$ measured by the AMS (Takeuchi and Ng, 2019;Nah et al., 2016). Here, the hydrolysis products ($C_{20}H_{33}NO_{6-12}$) make up less than 1% of their corresponding dinitrate species $C_{20}H_{32}N_2O_{8-14}$, demonstrating that hydrolysis of the main dinitrate dimers is not a significant loss term for this system. The lack of hydrolysis could come from the lack of water in the particles, which could differ from other experiments that have used seed aerosols (Takeuchi and Ng, 2019; Nah et al., 2016).

**3.5 Possible Oxidation Pathways**

In order to have oxidation reactions occurring in the particle-phase, such as the observed $C_{20}H_{32}N_2O_8$ to $C_{20}H_{32}N_2O_9$ conversion, there needs to be an oxidant or radical present with which to react. Radicals may act as oxidants through hydrogen abstraction reactions, with subsequent $O_2$ addition to the molecule that has lost a hydrogen. This would be a chemical pathway to form higher oxygenated molecules, and is analogous to similar reactions occurring in the gas phase (Molteni et al., 2019;Bianchi et al., 2019;Molteni et al., 2018). However, these reactions are not well-established in the particle-phase. Below, we speculate about some potential pathways and sources of radicals in the particle-phase.

Radical chemistry could be initiated from organic peroxides or peroxy nitrates incorporated in the particle phase, initiated from the scission of O-O bonds in organic peroxides. Considering the peroxy nitrate content is controlled by the concentration of $NO_2$ we would expect that this would only be important for experiments 1 and 3, and not for experiment 2.

Because the magnitude of oxidation is similar between experiments 2 and 3, peroxy nitrates are likely not an important source

of radicals. Degradation of dimers with a peroxy linkage could be prone to this effect, which would likely affect those species

formed in the gas phase, $C_{20}H_{32}N_2O_{8-10}$, and these molecules are also abundant in all experiments. For the higher oxygenated

molecules ($C_{20}H_{32}N_2O_{9,10}$), this effect would be obscured by the formation that dominates their time series, while for

$C_{20}H_{32}N_2O_8$ to disentangle this effect from evaporation becomes difficult.

The FIGAERO-CIMS also observes a small fraction (<0.1 µg m$^{-3}$) of $N_2O_5$ present in the particle phase, which could

act as an oxidant. The fraction of $N_2O_5$ in the particle phase decreases with time during the experiment (~50% over ~3 hours)

(shown in Figure S7), indicating it is being consumed or evaporating from the particle phase. Though, $N_2O_5$ is present in the

particle phase in experiments both with and without excess of $N_2O_5$, which is surprising considering its high volatility.

Overall, $N_2O_5$ present in the particle phase or organic peroxides could be responsible for the continued oxidation in

the particle phase. However, in the system presented here, we are unable to assess which potential oxidant or reaction is more

important because each oxidant will likely result in a similar reaction scheme and similar products.

**4 Conclusions and Atmospheric Implications**

The composition of $NO_3$-derived α-pinene SOA is dominated by dimers formed through $RO_2$-$RO_2$ reactions under the

experiment conditions in this study. Pye et al. (2015) modeled the atmospheric conditions during nighttime chemistry in the

southeastern US showing that the reactivity of $RO_2$ radicals in the atmosphere will react with either other $RO_2$ radicals (~40%)

or $HO_2$ radicals (60%), with little reactivity with $NO_3$ radicals. In pristine areas, the termination of $RO_2$ radicals can be

dominated by reactions with other $RO_2$'s (Yan et al., 2016). Given the prevalence of dimers formed from $NO_3$ chemistry

(regardless of the concentration of $NO_3$ radicals), as shown in our study, they will be atmospherically important and form in

most environments due to the relatively fast reaction rates between $RO_2$ radicals. These results illustrate that after formation

in the gas phase, dimers will condense to the particle phase where further reactions will proceed. Oxidation reactions are the

dominant reaction pathway accounting for 60-70% of the change in composition, while fragmentation (~10-20%) and dimer

decay to monomers (10-15%) are the minor pathways. Although there are changes in the nitrogen content, they do not appear

to be associated with hydrolysis of nitrate functional groups, but rather the repartitioning of low-volatility/semi-volatile highly

nitrated molecules. An open question remains the source of oxidation in the particle phase, as it could come from the

degradation of organic peroxides or from the small amount of $N_2O_5$ present. Despite not knowing the source of oxidation, the

results presented here, along with those shown for α-pinene derived SOA (from $O_3$) (Pospisilova et al., 2020;D'Ambro et al., 2018), demonstrate that SOA for both $NO_3$ or $O_3$, two of the major oxidants in the atmosphere, steadily evolves in the dark without external stimuli.

Performing the experiments in a way to balance the radical chemistry, via the addition of $HO_2$, would be ideal to replicate atmospheric chemistry accurately depending on the region. The results presented here are clearly tilted toward $RO_2$

– $RO_2$ reactions or $RO_2$-$NO_3$ reactions. The presence of $HO_2$ radicals will promote the formation of monomer hydroperoxide containing molecules over dimers, which will impact the ability to form lower volatility dimers, and increase the prevalence of hydroperoxide functional groups. The presence of hydroperoxide functional groups could promote particle-phase reactions if the peroxide groups are able to degrade and act as a source of radicals in the particle phase. Overall, particle-phase processing of $NO_3$ derived SOA is likely important regardless of the radical regime, since $RO_2 + RO_2$ dimer formation is always an

important sink of $RO_2$ radicals.

The overall effect of chemical aging of $NO_3$ derived SOA in the dark will be towards less volatile particles as a function of residence time in the atmosphere. The extent of the oxidation and aging in the particle phase could continue as long as there is a source of oxidant present. The reactions taking place in the particle phase result in changes of 25-30% (lower estimate) of the composition of $NO_3$ derived SOA. The particle-phase reactions slow / cease after 3 hours. In the atmosphere

there will be continual production and partitioning of oxidation products, which could continue this process throughout the identicnight.

The atmospheric consequence of these results is that we will typically over-predict the volatility of $NO_3$-derived SOA when aging reactions are not included, under-predict its impact on the lifetime of SOA in the atmosphere, and under-predict the lifetime of organonitrates in the atmosphere. Investigating the detailed chemistry of $NO_3$-derived SOA as a function of

relative humidity would provide further insight to assess the potential role of hydrolysis of nitrate functional groups, and its impact on these particle-phase processes.

**Data Availability**

The datasets are available upon request from the correspond authors. Chamber data will available upon publication from: https://data.eurochamp.org/

**Author Contributions**

DMB, CW, CM designed the study. Chamber experiments were carried out by DMB, CW, ELG, AB, SG, CM. Data analysis and interpretation was performed by DMB, CW, ELG, IR, JS, CM. DMB wrote the manuscript, with input from all co-authors. All co-authors read and commented on the manuscript.

**Competing interests**

The authors declare that they have no conflict of interest.

**Acknowledgements**

This work was supported by the Swiss National Science Foundation (grant 200020_172602, grant 200021_169787) as well as the European Union's Horizon 2020 research and innovation program through the EUROCHAMP-2020 Infrastructure Activity under grant agreement no. 730997. We would also like to thank Rene Richter for his assistance in installing and assembling
the experimental setup.

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

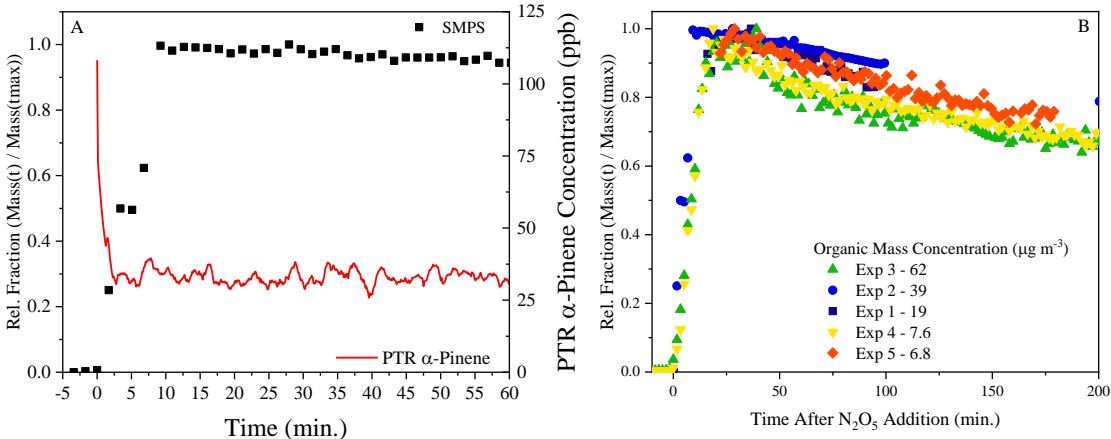

**Figure 1:** Evolution of precursor and SOA mass. **(a)** Example from experiment 2 showing prompt SOA formation and consumption of α-pinene. **(b)** All experiments performed showing the evaporation occurring during dark aging as measured by the SMPS, with the organic aerosol mass concentration associated with each experiment.

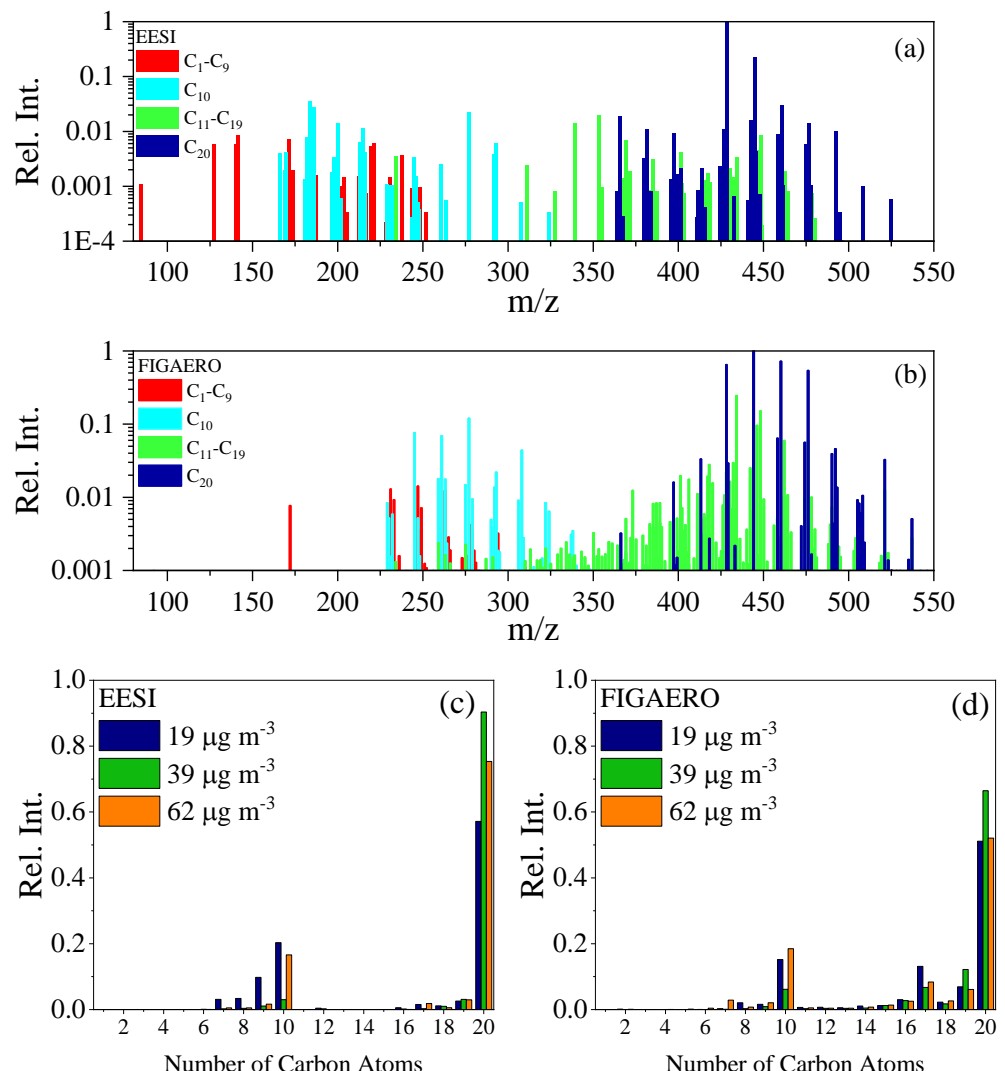

**Figure 2**: (**a**) Average mass spectrum presented for the EESI-ToF. (**b**) the first 20 min and FIGAERO-CIMS from the first desorption from experiments 1 (blue), 2 (green), 3 (orange), respectively, taking place between the first 10-46 min of the experiment. The main set of molecules correspond to $C_{20}H_{32}N_2O_x$ (x = 8 – 13) for both instruments. (**c & d**) Binned carbon distribution for experiments 1 – 3 for the same time period for the EESI-ToF and FIGAERO, respectively.

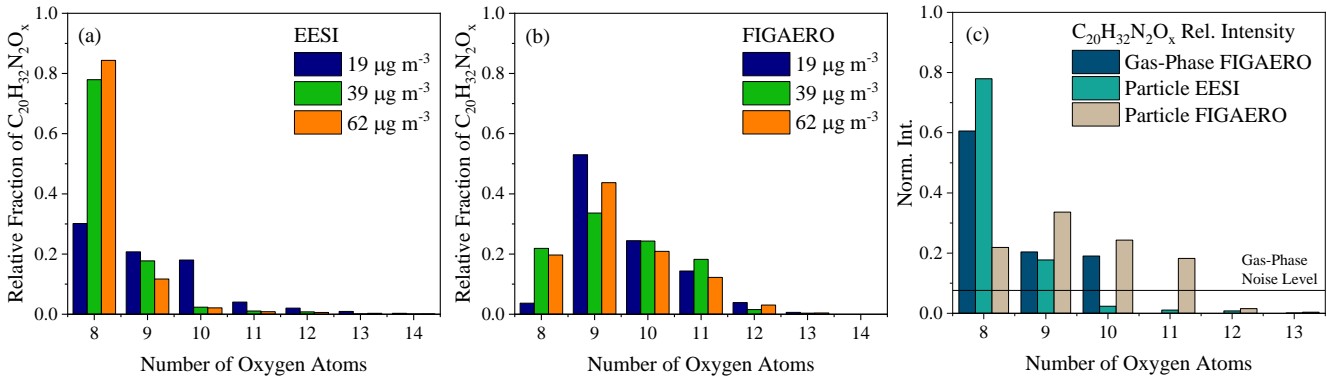

**Figure 3: (a & b)** Oxygen atom distribution for $C_{20}H_{32}N_2O_{8-14}$ molecules observed by both the EESI-ToF and FIGAERO-CIMS, respectively, experiments 1 (blue), 2 (green), 3 (orange), normalized to the total $C_{20}H_{32}N_2O_x$ signal. The EESI-ToF data is averaged over the first 10 min of the experiments while the FIGAERO-CIMS data represents the first filter desorption. **(c)** Oxygen atom distribution for the FIGAERO-CIMS (gas phase) averaged over the first ~5 min from experiment 2. The solid

line denotes the limit of detection for the FIGAERO-CIMS.

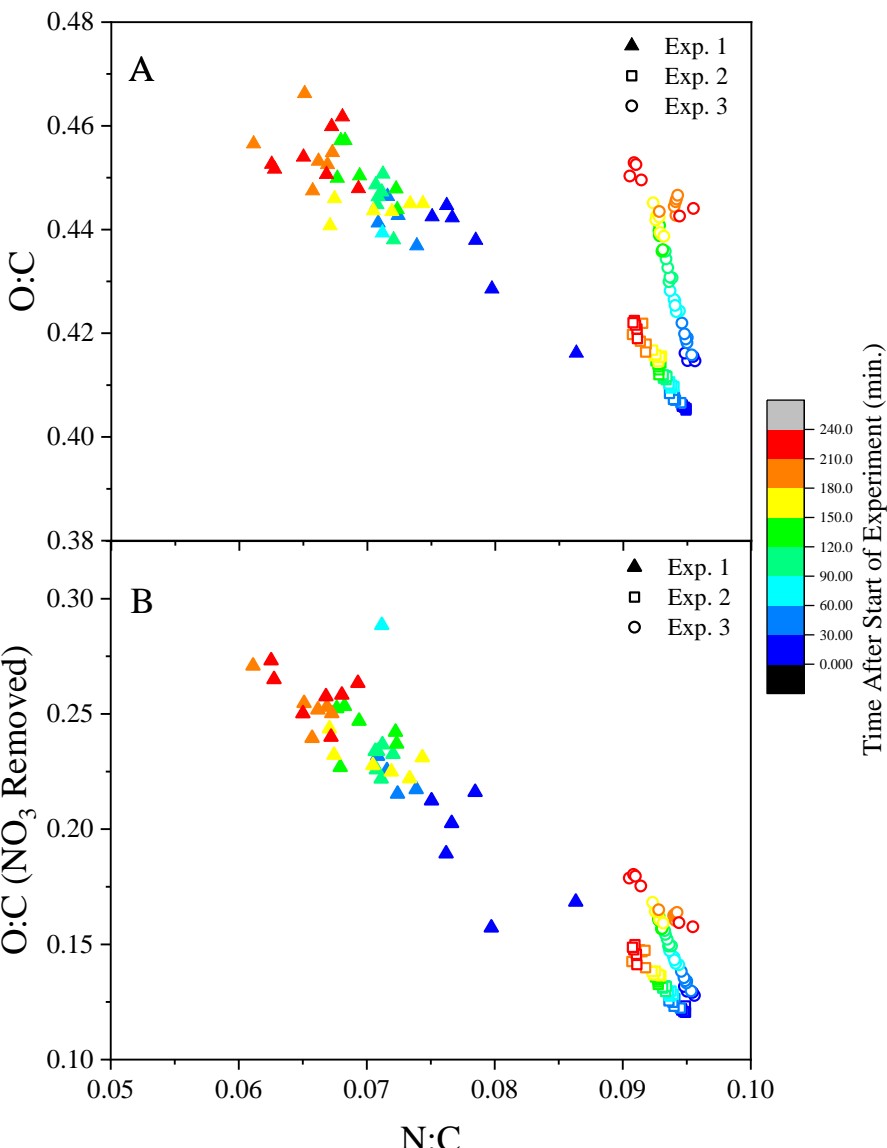

**Figure 4: (a)** N:C and O:C ratios measured by the EESI-ToF over the course of the experiment (color scale). **(b)** O:C ratio altered by the removal of –ONO$_2$ groups from the O:C ratio where #O$_{non-NO3}$ = #O$_{total}$ – 3*#N.

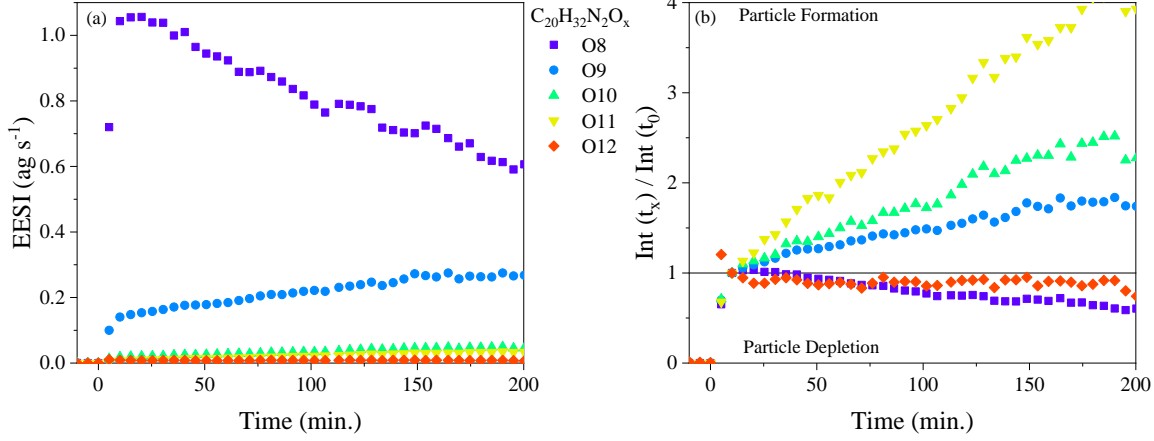

**Figure 5: (a)** Wall-loss corrected time series of the $C_{20}H_{32}N_2O_{8-13}$ molecules observed by the EESI-ToF over the course of

dark aging in the chamber from exp 2. **(b)** All signals normalized to their intensity at $t = 10$ min.

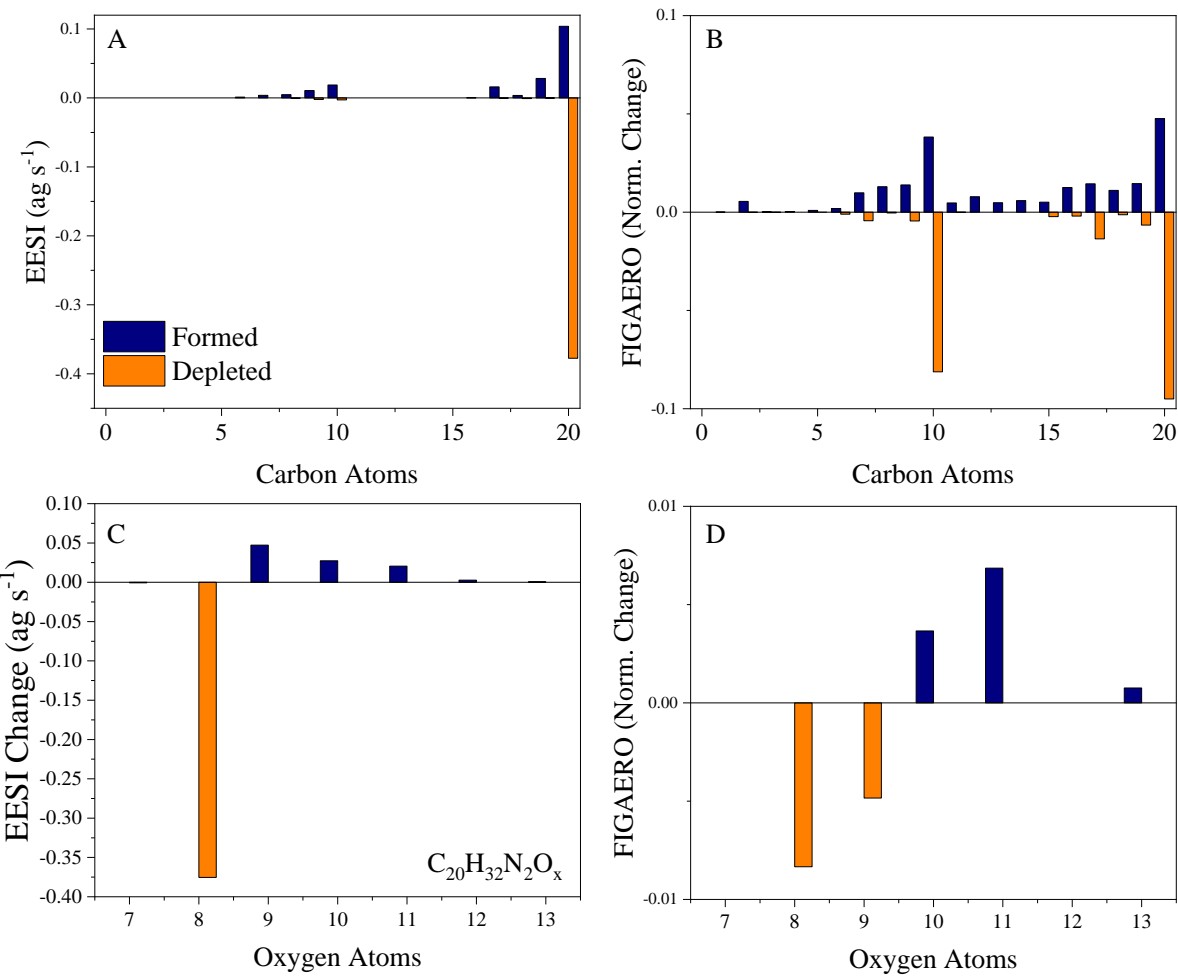

**Figure 6: (a & b)** Carbon distribution of the molecules formed (blue) and depleted (orange) in the particle phase during aging for both the EESI-ToF **(a)** and FIGAERO-CIMS **(b)**, respectively for exp 3. **(c & d)** The change in the oxygen distribution for

the $C_{20}H_{32}N_2O_{8-13}$ molecules during dark aging for the EESI-ToF **(c)** and FIGAERO-CIMS **(d)**, respectively. For panels A and C , a difference is obtained by taking the difference from $t = 15$ min and $t = 150$ min from the EESI-ToF. Panels B and D are obtained by taking the difference in the relative sensitivity from the 1$^{st}$ and 3$^{rd}$ filter.  For all figures, the formation is shifted negative on the x-axis relative to the nominal carbon number, and the depletion is shifted positive on the x-axis comparatively.

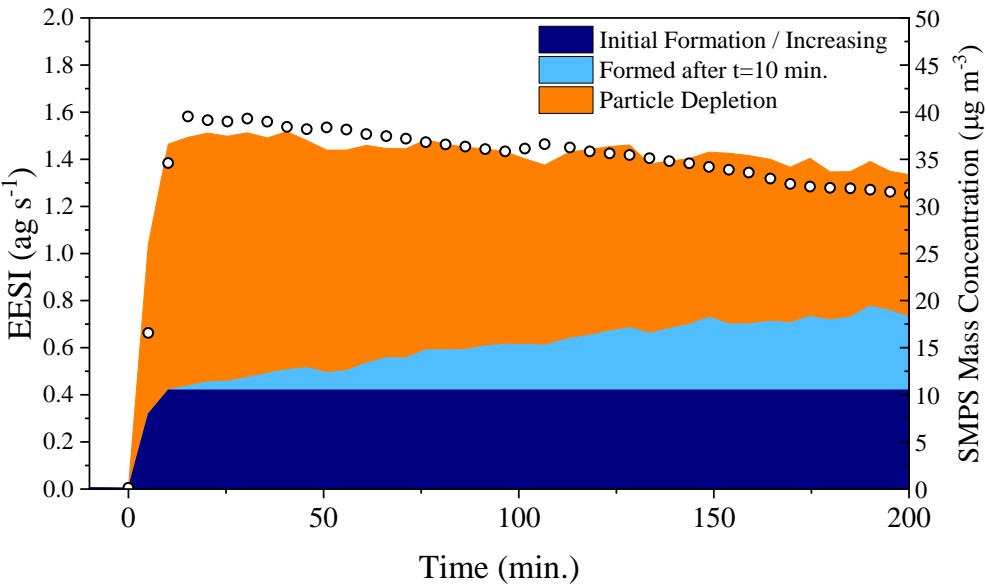

675

**Figure 7: (left axis)** Total EESI-ToF intensity plotted as a function of the contribution from different sources, particle phase

formation products determined from those molecular formulae that increase during dark aging after $t$ = 10 min from experiment

2. **(right axis)** Measured SMPS mass concentration corrected for wall losses.