# Peer review of "Particle-phase processing of $\alpha$ -pinene $\text{NO}_3$ secondary organic aerosol in the dark"

_Atmospheric Chemistry and Physics, 2021_

## Referee Comment (RC1)

Review of Bell et al ACP 2021

This paper presents a new set of chamber experiments exploring the condensed-phase dark aging of NO3 + a-pinene SOA, which chemically speciated observations (EESI, FIGAERO-CIMS) illuminating the continued oxidation of the SOA. The analysis could be expanded to increase the impact of this paper, and I recommend additional analysis be included in the main body of the paper.

Major comments/suggestions:
1) The authors use the MCM model to determine the most likely RO2 bimolecular reaction partner, and then note that the RO2+RO2 product channel is apparently more dominant based on observed products. Based on this empirical observation, could you use the model to infer what the RO2+RO2 rate constant must be, for these C10 nitrate functionalized RO2's? It seems to me this is an opportunity your data give you that should be exploited! Suggest to include an additional section on modeling RO2 fate.
2) I don't understand the claim that peroxynitrates are controlled by RO2 + NO3 reactions. I think they would be controlled by nitrato-RO2 + NO2 reactions, which makes one of your ideas about oxidant sources less sound.
3) You discuss fragmentation as yielding only CH2O as volatile fragments, but the loss of organonitrates that cannot be explained by hydrolysis to HNO3 suggests that there must also be some fragmentation to high-volatility organonitrates. This would be good to elaborate upon. Would these be detectable in any of your gas-phase measurements?
4) Could semivolatiles be repartitioning differentially to the chamber walls? Your discussion of wall losses seems to assume a consistent loss rate for all species, but these rates could be species-dependent. Could the apparent loss of O8 be due to greater wall repartitioning for that molecule than for higher-oxidized, heavier molecules? I suggest thinking about the speciated wall partitioning discussed in Krechmer et al. 2020 (https://pubs.acs.org/doi/abs/10.1021/acs.est.0c03381)
5) Sections 3.2 and 3.3 have the same title
6) Around line 155: can dimers also form in the EESI, in the reverse of the fragmentation you discuss?

Minor / technical points:
1) Line 11 "in or downwind of polluted"
2) Line 15: "in the absence of external stimuli" feels a bit vague to me -- what you really mean is aging in the dark, right? This phrase is also used later. Consider rewording? But this is a style choice, so just a suggestion to think about
3) Line 29: 'Unlike isoprene, monoterpenes are emitted"
4) Line 30: "and NO3) play an important"
5) Line 50: "still not well understood"
6) Line 59: "determine the absolute scale" is unclear to me. Perhaps something like "determine the magnitude of the effect"?

7) Section 2.1: find a place here to mention that these experiments were run in batch mode; also suggest to spell out EESI and FIGAERO here too (even though they came also in the abstract), since you spell out other acronyms in this section
8) Line 67: which instrument is the thermo-denuder in front of?
9) Line 75: indicate the approximate concentration of the cresol contaminant?
10) Line 83: "cresol contamination constitute ~1-2%"
11) Line 86: "experiments 1 - 3, ~100 ppb"
12) Table 1 caption: I suggest to add some rate constant modeling info to this caption: "based on a fit to the VOC decay, assuming XXX as the NO3 + apin rate constant at XX C"
13) Eq 1 formatting: need a space between equation and label
14) Line 117: "sampling the aerosol at 1 L min-1"
15) Line 122 "ion guides and were separated" -- in general, check for verb tense consistency: most are past tense, but some a present tense.
16) Lin 127: "subtracting the background filter periods from the adjacent chamber sampling periods. A filtering"
17) Around line 132: this equation needs a number. There is some repeated text before and after "Avogardo's number and a conversion factor…" - remove one
18) Line 135 "including particle size,"
19) Line 140: "and had low sensitivity"
20) Line 143: give formula for acetonitrile at its first instance, to help reader interpret later clusters you mention
21) Line 146; " for all C20H32N2Ox molecules" -- if this is in fact what you mean?
22) Line 175: "Tofware"
23) Line 177: "For the first filter in Exp. 2, due to a software failure, the filter was stored wrapped in aluminum foil for ~7 H after … was done prior to desorption."
24) Line 178: " were similar to other experiments."
25) Line 187: "consumed, because the injection of N2O5 was less"
26) Line 199: " observations (Takeuchi and Ng, 2019), and with RO2+ RO2 reactions."
27) Line 201: "dimers over monomers"
28) Around line 206: I don't understand how peroxynitrates would be formed from RO2 + NO3 reactions. WOuldn't they be formed by nitrato-RO2 + NO2 reactions?
29) Lines 222-224: The first sentence of this section doesn't make sense to me, sharpen / reword / make more specific?
30) Line 226; "larger SOA yields were observed under an RO2 + RO2 dominant"
31) …. Rest of section: make every instance consistently "RO2 + X" - currently some have the + sign and some have long dashes --
32) Line 236: "HO2 is not an important RO2 reaction partner since there"
33) Line 240:" peroxy linkage. This molecule is the dominant dimer"
34) Around line 257: See comment above about expanding interpretation of RO2 + RO2 rate constant based on your observations.
35) Line 261; "and the FIGAERO-CIMS, were used to"
36) Line 265 spelling FIGAERO

37) Line 282: units "ag s-1 h-1" don't make sense to me. Per second and per hour? (on next line too)
38) Line 312: spurious comma at the end of the line
39) Line 331: reorder confusing sentence: "Approximately half of the total depletion observed arises from an increase in oxidation, with the remainder coming from evaporative losses."
40) Line 343-345: this last line of the paragraph, about no specific loss of -ONO2 groups, is confusing to me.
41) Line 361: "initiated from the scission of O-O bonds in organic peroxides"
42) Next line: as mentioned above, I don't see why PANs concentration would be drive by [NO3] (rather, I would expect a dependence on [NO2])
43) Around line 371-372: could the same N2O5 measurement be an artefact / wall background?
44) Line 373: "phase or organic peroxides could be"
45) Line 380: remove "making up the difference"
46) Line 384: "in the gas phase, dimers will"
47) Around linke 387: doesn't this say fragmentation isn't just CH2O?
48) Line 389: 'organic peroxides or from"
49) Line 390: "presented here, along with… ), demonstrate that"
50) Line 397: "Overall, particle-phase … regime, since .. is always an important sink of RO2"
51) Line 405: suggest to start new paragraph with this sentence and edit to:" The atmospheric consequence of these results is that we will typically over-predict…"
52) Figure 1: what are the numbers after the experiment numbers in panel B caption?
53) Figure 2 caption references to panels a and b are confusing, reorder text?
54) Figure 4 notation about #O is confusing. What about "#O(non-NO3) = #O(total) - 3*#N"?

---

## Author Comment (AC1)

The Reviewer comments are in black, the replies are provided in blue, and the changed text in the manuscript are shown in orange.

Review of Bell et al ACP 2021
This paper presents a new set of chamber experiments exploring the condensed-phase dark aging of NO3 + a-pinene SOA, which chemically speciated observations (EESI, FIGAERO-CIMS) illuminating the continued oxidation of the SOA. The analysis could be expanded to increase the impact of this paper, and I recommend additional analysis be included in the main body of the paper.

We would like to thank reviewer for their constructive comments and questions.

Major comments/suggestions:
1) The authors use the MCM model to determine the most likely RO2 bimolecular reaction partner, and then note that the RO2+RO2 product channel is apparently more dominant based on observed products. Based on this empirical observation, could you use the model to infer what the RO2+RO2 rate constant must be, for these C10 nitrate functionalized RO2's? It seems to me this is an opportunity your data give you that should be exploited! Suggest to include an additional section on modeling RO2 fate.

We believe that the data here although useful, is not the whole story needed to be able to constrain the values for RO2+RO2 rate constants. In order to better constrain these rate constants, measurements about radical concentrations and well-calibrated gas-phase measurements would be required.

2) I don't understand the claim that peroxynitrates are controlled by RO2 + NO3 reactions. I think they would be controlled by nitrato-RO2 + NO2 reactions, which makes one of your ideas about oxidant sources less sound.

We would like to thank the reviewer for this comment. We should have said RO2 + NO2 reactions, as RO2 + NO3 will react to form an alkoxy (RO) radical. Whenever we have large concentrations of NO3 there are corresponding large concentrations of NO2 to go along with it.

This was changed on line 208 to say with the highlighted changes in orange: "Monomer dinitrates likely form via an $RO_2$ + **$NO_2$** reaction to form peroxynitrate functional groups (Chan et al., 2010), which is speculated to form monomer trinitrates in the isoprene + $NO_3$ system (Zhao et al., 2020). The lack of their formation in experiment 2 agrees with the smaller concentrations of $N_2O_5$ injected, and corresponding smaller amount of $NO_2$ present."

3) You discuss fragmentation as yielding only CH2O as volatile fragments, but the loss of organonitrates that cannot be explained by hydrolysis to HNO3 suggests that there must also be some fragmentation to high-volatility organonitrates. This would be good to elaborate upon. Would these be detectable in any of your gas-phase measurements?

Thank you for this comment. In the gas-phase, we observed a few species that increased more than 25% during the course of dark aging, which would be an indicator for the formation and subsequent evaporation of high volatility organo-nitrates. The molecules increasing include: $C_2H_4O_3$, $C_3H_6O_3$, $C_4H_8O_3$, $C_3H_5NO_5$, $C_4H_7NO_5$, $C_5H_9NO_5$. The use of CH2O was an example because the loss of a single carbon containing molecules is the most significant loss pathway.

We added the following on line 365:

"We should note, we do observe a few molecules increasing more than 25% in the gas-phase consistent with fragmentation reactions, including: C2H4O3, C3H6O3, C4H8O3, C3H5NO5, C4H7NO5, C5H9NO5"

4) Could semivolatiles be repartitioning differentially to the chamber walls? Your discussion of wall losses seems to assume a consistent loss rate for all species, but these rates could be species-dependent. Could the apparent loss of O8 be due to greater wall repartitioning for that molecule than for higher-oxidized, heavier molecules? I suggest thinking about the speciated wall partitioning discussed in Krechmer et al. 2020 (https://pubs.acs.org/doi/abs/10.1021/acs.est.0c03381)

Although the loss of organo-nitrates doesn't necessarily mean there must be a fragmentation pathway to form high-volatility organo-nitrates, we agree with the reviewer suggestion that it could result from semi-volatile repartitioning to the chamber walls (Krechmer et al. 2020 and (Bertrand et al., 2018)). These molecules are generally thought to be non-volatile in nature (Mohr et al., 2019). However, when performing isothermal evaporation experiments in the atmospheric simulation chamber both the $C_{20}H_{32}N_2O_{8,9}$ are susceptible to repartitioning, giving credence to the idea these molecules are repartitioning directly.

Figure S8 was added to the Supplement:

[Figure]

**Figure S8:** After experiment 3, clean air was injected into the chamber at 100 L min$^{-1}$ for 60 min. The time traces here correspond to the EESI-ToF intensity relative to the time at the beginning of the injection of clean air into the chamber for the series of $C_{20}H_{32}N_2O_{8-12}$.

We have added the following section to address the evaporation of semi-volatile dimers on line 373:

"Figure 1b shows that the SOA evaporates during aging in the chamber. Evaporative loss of dinitrate dimers is possible, though not common considering dimers typically have very low volatility. However, when performing isothermal evaporation measurements in the atmospheric simulation chamber both the $C_{20}H_{32}N_2O_{8,9}$ are susceptible to repartitioning (Figure S8). Because these dimers are low-volatility to semi-volatile molecules they are in equilibrium between the particles, gas-phase, and the walls (Bertrand et al., 2018;Krechmer et al., 2020). Therefore, part of the loss of $C_{20}H_{32}N_2O_8$ (and O9?) is due to repartitioning and accounts for a significant fraction of the organonitrates lost from the particle phase observed by the EESI-ToF, and a significant fraction of the decreasing signal in the FIGAERO-CIMS.»

We have changed part of our conclusions (Line 417) section to:

"Although there are changes in the nitrogen content, they do not appear to be associated with hydrolysis of nitrate functional groups, but rather the repartitioning of low-volatility/semi-volatile highly nitrated molecules."

5) Sections 3.2 and 3.3 have the same title

Thank you for pointing this out, the title has been changed to: "Evolution in the Particle Phase Composition"

6) Around line 155: can dimers also form in the EESI, in the reverse of the fragmentation you discuss?

It is possible for dimers to form in the EESI, but this is typically the case when there are very high concentrations of specific individual species. For instance, when performing calibrations with levoglucosan, dimers ($Na^+$-Levoglucosan-levoglucosan) are observed with a relative intensity of 0.01 – 1% of the of the typical adduct $Na^+$-Levoglucosan under our ion-guide settings. Similar to the ACN clustering discussed, the prevalence of these "dimers" are highly dependent on the ion-guide settings. Overall, these "dimers" are highly non-linear with aerosol mass concentration, and generally are not significant.

Minor / technical points:

Thank you for the minor points, all of them have been addressed either by changing the main text with the suggested change (i.e. done) or through the above comments.

1) Line 11 "in or downwind of polluted" done

2) Line 15: "in the absence of external stimuli" feels a bit vague to me -- what you really mean is aging in the dark, right? This phrase is also used later. Consider rewording? But this is a style choice, so just a suggestion to think about good point here, it was changed to "in the dark"

3) Line 29: ' Unlike isoprene, monoterpenes are emitted" done

4) Line 30: "and NO3) play an important" done

5) Line 50: "still not well understood" done

6) Line 59: "determine the absolute scale" is unclear to me. Perhaps something like "determine the magnitude of the effect"? done

7) Section 2.1: find a place here to mention that these experiments were run in batch mode; also suggest to spell out EESI and FIGAERO here too (even though they came also in the abstract), since you spell out other acronyms in this section done

Line 64: "A series of batch-mode chamber experiments (Table 1) investigating formation…"
And added the description of the acronyms.

8) Line 67: which instrument is the thermo-denuder in front of?
Line 69 ", a thermo-denuder coupled with an SMPS, …"

9) Line 75: indicate the approximate concentration of the cresol contaminant? done

10) Line 83: "cresol contamination constitute ~1-2%" done

11) Line 86: "experiments 1 - 3, ~100 ppb" done

12) Table 1 caption: I suggest to add some rate constant modeling info to this caption: "based on a fit to the VOC decay, assuming XXX as the NO3 + apinrate constant at XX C" done

13) Eq 1 formatting: need a space between equation and label done

14) Line 117: "sampling the aerosol at 1 L min-1" done

15) Line 122 "ion guides and were separated" -- in general, check for verb tense consistency: most are past tense, but some a present tense. done

16) Line 127: "subtracting the background filter periods from the adjacent chamber sampling periods. A filtering" done

17) Around line 132: this equation needs a number. There is some repeated text before and after "Avogadro's number and a conversion factor…" - remove one done

18) Line 135 "including particle size," done

19) Line 140: "and had low sensitivity" done

20) Line 143: give formula for acetonitrile at its first instance, to help reader interpret later clusters you mention done

21) Line 146; " for all C20H32N2Ox molecules" -- if this is in fact what you mean? Done, and yes

22) Line 175: "Tofware" done

23) Line 177: "For the first filter in Exp. 2, due to a software failure, the filter was stored wrapped in aluminum foil for ~7 H after … was done prior to desorption." Adapted this suggestion to: "For the first filter in Exp. 2, due to a software failure, the filter was stored wrapped in aluminum foil for ~7 h before being desorbed."

24) Line 178: " were similar to other experiments." Done

25) Line 187: "consumed, because the injection of N2O5 was less" Done

26) Line 199: " observations (Takeuchi and Ng, 2019), and with RO2+ RO2 reactions." Done

27) Line 201: "dimers over monomers" done

28) Around line 206: I don't understand how peroxynitrates would be formed from RO2 +NO3 reactions. Wouldn't they be formed by nitrato-RO2 + NO2 reactions? This was changed in accordance with the major comment above.

29) Lines 222-224: The first sentence of this section doesn't make sense to me, sharpen / reword / make more specific?

"Because particle-phase reactions are not well understood, it is necessary to understand what radical pathways control the initial composition of the SOA."

30) Line 226; "larger SOA yields were observed under an RO2 + RO2 dominant" done

31)…. Rest of section: make every instance consistently "RO2 + X" - currently some have the + sign and some have long dashes – done

32) Line 236: "HO2 is not an important RO2 reaction partner since there" done

33) Line 240:" peroxy linkage. This molecule is the dominant dimer" done

34) Around line 257: See comment above about expanding interpretation of RO2 + RO2 rate constant based on your observations. Talked about this above.

35) Line 261; "and the FIGAERO-CIMS, were used to" done

36) Line 265 spelling FIGAERO done

37) Line 282: units "ag s-1 h-1" don't make sense to me. Per second and per hour? (on next line too)

we modelled the decay rate and fit the decay with an exponential fit and now report a decay rate of " 0.01 min$^{-1}$ "

38) Line 312: spurious comma at the end of the line done

39) Line 331: reorder confusing sentence: "Approximately half of the total depletion observed arises from an increase in oxidation, with the remainder coming from evaporative losses." done

40) Line 343-345: this last line of the paragraph, about no specific loss of -ONO2 groups, is confusing to me. Removed this sentence since it was repeating the point of the next paragraph.

41) Line 361: "initiated from the scission of O-O bonds in organic peroxides" done

42) Next line: as mentioned above, I don't see why PANs concentration would be drive by [NO3] (rather, I would expect a dependence on [NO2]) changed this to NO2 in the text, in line with the comment above.

43) Around line 371-372: could the same N2O5 measurement be an artefact / wall background?

The FIGAERO-CIMS measured directly gas-phase $N_2O_5$ and semi-simultaneously particle-phase N2O5 via heating collected filter samples (details see Methods). The gas-phase N2O5 signal increased from a very low background with adding N2O5 into the chamber, and decreased very fast back to a very low level in a few minutes. Thus, gas-phase N2O5 wouldn't impact the further particle-phase measurement. The particle-phase N2O5 from the filters had thermograms consistent with those of other species during heating round. This included starting near zero at the beginning of heating and increasing with heating temperature, and had a maximal desorption temperature of 80-90°C, and decreased to almost zero at the end of the heating cycle. This indicates that the N2O5 signal was not an artefact from surfaces of the instrument.

44) Line 373: "phase or organic peroxides could be" done

45) Line 380: remove "making up the difference" done

46) Line 384: "in the gas phase, dimers will" done

47) Around line 387: doesn't this say fragmentation isn't just CH2O? addressed in above comment

48) Line 389: ' organic peroxides or from" done

49) Line 390: "presented here, along with… ), demonstrate that" done

50) Line 397: "Overall, particle-phase … regime, since .. is always an important sink of RO2" done

51) Line 405: suggest to start new paragraph with this sentence and edit to:" The atmospheric consequence of these results is that we will typically over-predict…" done

52) Figure 1: what are the numbers after the experiment numbers in panel B caption?
We added to the figure caption " with the organic aerosol mass concentration associated with each experiment."

53) Figure 2 caption references to panels a and b are confusing, reorder text?
We reordered the text to clarify the EESI is (a) and FIGAERO is (b)

54) Figure 4 notation about #O is confusing. What about "#O(non-NO3) = #O(total) - 3*#N"?

We changed the Figure caption accordingly.

Bertrand, A., Stefenelli, G., Pieber, S. M., Bruns, E. A., Temime-Roussel, B., Slowik, J. G., Wortham, H., Prévôt, A. S. H., El Haddad, I., and Marchand, N.: Influence of the vapor wall loss on the degradation rate constants in chamber experiments of levoglucosan and other biomass burning markers, Atmos. Chem. Phys., 18, 10915-10930, 10.5194/acp-18-10915-2018, 2018.

Chan, A. W. H., Chan, M. N., Surratt, J. D., Chhabra, P. S., Loza, C. L., Crounse, J. D., Yee, L. D., Flagan, R. C., Wennberg, P. O., and Seinfeld, J. H.: Role of aldehyde chemistry and $NO_x$ concentrations in secondary organic aerosol formation, Atmos. Chem. Phys., 10, 7169-7188, 10.5194/acp-10-7169-2010, 2010.

Krechmer, J. E., Day, D. A., and Jimenez, J. L.: Always Lost but Never Forgotten: Gas-Phase Wall Losses Are Important in All Teflon Environmental Chambers, Environmental Science & Technology, 54, 12890-12897, 10.1021/acs.est.0c03381, 2020.

Mohr, C., Thornton, J. A., Heitto, A., Lopez-Hilfiker, F. D., Lutz, A., Riipinen, I., Hong, J., Donahue, N. M., Hallquist, M., Petäjä, T., Kulmala, M., and Yli-Juuti, T.: Molecular identification of organic vapors driving atmospheric nanoparticle growth, Nat. Commun., 10, 4442, 10.1038/s41467-019-12473-2, 2019.

Zhao, D., Pullinen, I., Fuchs, H., Schrade, S., Wu, R., Acir, I. H., Tillmann, R., Rohrer, F., Wildt, J., Guo, Y., Kiendler-Scharr, A., Wahner, A., Kang, S., Vereecken, L., and Mentel, T. F.: Highly oxygenated organic molecules (HOM) formation in the isoprene oxidation by NO3 radical, Atmos. Chem. Phys. Discuss., 2020, 1-28, 10.5194/acp-2020-1178, 2020.

---

## Author Comment (AC2)

**Reviewer #2**

This manuscript (which is a companion paper to one submitted by Wu et al.) describes results of a laboratory study of the effect of aging in the dark on the mass and composition of SOA formed from the reaction of NO3 radicals with a-pinene for a few different concentrations of a-pinene and N2O5, which was the source of NO3 radicals. Experiments were conducted in a Teflon chamber, SOA mass and size were monitored with an SMPS, and gas and particle composition were monitored with a FIGAERO-CIMS and EESI-TOF. The observations are thoroughly discussed, and various possible explanations, such as evaporation, oxidation, and monomer-dimer reactions are proposed. In general, however, given the complexity of the system, the lack of information on the molecular structures of the SOA components (only elemental formulas are available), and the non-quantitative MS analyses, it was not possible to draw convincing conclusions about the physical or chemical processes that might have altered the SOA in the dark. Nonetheless, the data set is interesting, and future studies may provide more detailed data that can help to explain the results. I think the manuscript can be published after the following comments are addressed.

We would like to thank the reviewer for their questions and comments.

Specific Comments

1. Line 205: The reaction RO2 + NO3 forms RO + NO2 + O2, not peroxynitrates (ROONO2). I assume you meant RO2 + NO2 –> ROONO2.

Thank you for this comment, we have changed the RO2 + NO3 to RO2 + NO2.

2. How do you propose that peroxynitrates are converted to nitrates? The only ROONO2 reactions I am aware of are reversible formation of RO2 + NO2 and decomposition to R(O) + HNO3. It seems more likely that the additional nitrates observed in the excess N2O5 experiments 1 and 3 are formed by reactions of alcohols with N2O5: ROH + N2O5 –> RONO2 + HNO3, which is a well-known reaction that is used to synthesize organic nitrates from the corresponding alcohols.

We assume the reviewer is referring to the fact that peroxynitrates are likely unstable and have a lifetime that is very short in the aerosol phase. Overall, we cannot distinguish the difference between organonitrates and peroxy nitrates with our current measurement techniques, so there is a possibility we observe molecules formed from both RO2 + NO2 and ROH + N2O5. Though, Zhao et al. (2020) suggests that the route via RO2 + NO2 is an important driver to form trinitrates in chamber experiments studying isoprene + NO3, which will also be a valid pathway in the experiments shown here. (Highlighted on line 210).

Once in the particle phase, we propose that peroxy nitrates can undergo unimolecular scission, which would create an NO3 radical and an alkoxy radical. After this condensed phase radical chemistry is not clear, at least this is true in bulk polymer chemistry (Smith et al., 2018). The suggested route to form a carbonyl is unlikely because the formation of C20H30NOx molecules is not significant, which would occur with the conversion of a nitrate to a carbonyl. But once radicals are formed, presumably cascading H-abstraction reactions will continue through the aerosol until finally terminating.

3. Line 255: Because the RO2 + RO2 and RO2 + NO3 reactions both lead to the same alkoxy radicals, and these can go on to form monomers that then form dimers in particles, an alternative explanation for the similarity in SOA dimer composition in the two radical regimes is that most of the dimers are formed in the particles and that gas-phase dimers are minor. Since these MS methods are not quantitative, it is not possible to draw conclusions on the importance of gas-phase dimers.

This comment is in stark contrast to reviewer #1, which suggests that we can use the current data to infer the rates of reactions of RO2+RO2 reactions. We agree with reviewer #2 in that we are unable to assess exact rates of reactions (e.g. rates of RO2 + RO2 reactions) for gas-phase dimer formation without well calibrated measurements of gas-phase concentrations.

Although the alkoxy pathway would be the same for both of the radical pathways (RO2 + RO2/NO3), the ultimate branching can be remarkably different between the two, since RO2+RO2 has other termination pathways (e.g dimers, alcohol, or carbonyl). Besides the difference in the branching pathways, the alkoxy pathway ($RO_2$ + $NO_3$) is predicted to proceed mostly to pinonaldehyde. Based on saturation vapor concentration estimates, this molecule would not participate significantly in the formation of SOA or the formation of dimers. In contrast to this, many recent studies have demonstrated the importance of RO2 + RO2 reactions to the formation of dimers. (Berndt et al., 2018a;Berndt et al., 2018b;Zhao et al., 2018;Ng et al., 2008;Molteni et al., 2019;Rissanen et al., 2015;Simon et al., 2020;Heinritzi et al., 2020)

Additionally, the idea that both reaction pathways could lead to the same products and ultimately be just as important in the formation of SOA does not agree with other results that show SOA yields are smaller for both isoprene + NO3 and α-pinene + NO3 when dominated by RO2 + NO3 chemistry(Bates et al., 2021;Ng et al., 2008).

Along these lines, we added to the manuscript beginning on line 263: "These findings are consistent with the importance of dimers formed from $RO_2$ + $RO_2$ reactions in other systems (Berndt et al., 2018a;Berndt et al., 2018b;Zhao et al., 2018;Ng et al., 2008;Molteni et al., 2019;Rissanen et al., 2015;Simon et al., 2020;Heinritzi et al., 2020), and with higher SOA yields from $NO_3$ initiated oxidation during dominant $RO_2$ + $RO_2$ chemistry (Ng et al., 2008;Bates et al., 2021)."

4. Line 329: Since the EESI is not calibrated, how can you measure a mass flux?

Lines 130 – 138 of the manuscript detail the conversion from counts $sec^{-1}$ to attogram $sec^{-1}$. The conversion is based on assuming that all species are detected equally well and converting the mass observed at the detector of the EESI-MS. A comparison of the measured ag $sec^{-1}$ against the measured mass by the SMPS is presented in Figure S2, and demonstrates there is a good agreement. We have added our levoglucosan calibration points to demonstrate the same linearity from these experiments.

5. Since neither the EESI-TOF or the FIGAERO-CIMS signals have been calibrated, the authors cannot assume that all compounds have the same sensitivity. This makes it difficult to draw conclusions from the changes observed in MS signals over time. For example, if reversible (non-oxidative) monomer exchange reactions were occurring in the particles to form dimers with different structures and detection sensitivities, then this could appear as oxidation when it is not. One can imagine a variety of such scenarios that confuse an interpretation of the MS observations.

The response of the EESI-ToF has been shown to vary by ~ 1 order of magnitude when comparing oxidation products from different types of volatile organic compounds (Wang et al., 2021). Also, the EESI agrees very well with predicted SOA composition and mass based on measured components in the gas-phase by the PTR-3 for other SOA systems (Surdu et al., 2021). Additionally, we demonstrate the good agreement between the measured mass by the SMPS and the corresponding EESI signal (ag $sec^{-1}$) for the experiments presented here. Together, these results suggest even if there are differences in sensitivity, the EESI is capturing the chemical evolution.

In the proposed scenario, the formation of reversible dimers comes from non-observed monomers (or observed in a limited way). In order for this scenario to result in agreement with the observed SMPS mass, then the formation / consumption would have to perfectly equal out because in the scenario the monomers are not observed. Also, both the EESI and the FIGAERO-CIMS would have to be biased

in the same exact way in order to observe similar processes without observed reversible (non-oxidative) monomer exchange reactions.

The following has been added to the text as a paragraph at line 316:

[revised manuscript text omitted]

---

## Author Comment (AC3)

Reviewer #3 (Nga Lee Ng)

I have a few quick comments for the authors to consider:

Thank you for adding a few questions and comments.

1. It appears that Experiments #1 and #3 were conducted under conditions that were almost identical (Table 1), why is the maximum SOA different by a factor of ~3? Same question applies to the data shown in the figures, how shall the results from Experiments #1 and #3 be explained and interpreted? Shall one expect the results to be comparable or different?

For the experiments we performed we attempted to be as reproducible as possible. However, one downside in doing injections of N2O5 from solid crystals comes from the short injection times (1-10sec.) due to the large concentration present from the solid. We attempted to mitigate this by mixing the chamber for a short period by injecting air (~100 L min$^{-1}$) along with the N2O5. However, this will create plume effects in the chamber which can produce inhomogeneities throughout the chamber. Because of these inhomogeneities, differences in the yields of formation are certainly possible. Though, the mass spectra presented in both Exp 1 and 3 are similar suggesting the overall reaction pathway is similar. Additionally, the radical domain (RO2+RO2 or RO2+NO3) does not impact the type of condensed phase processes occurring (nor their magnitude). We believe the observations are robust despite the clear differences.

2. Line 42-45: For the sentence "When experiments were conducted in the dark….", it might be better to separate this sentence into two to avoid confusion, as Takeuchi and Ng did not use the changes in elemental ratios (N:C and O:C) to evaluate organic nitrate hydrolysis (NO3,org/Org is used as a proxy to infer hydrolysis in Takeuchi and Ng).

Thank you for this comment, we tried to clarify this sentence by changing it to (Line 42):

"When experiments were conducted in the dark, monoterpene + NO$_3$ products were shown to steadily evolve, with a steady change in O:C and N:C ratios as reported by Nah et al. (2016). In Takeuchi and Ng (2019) ~9-17% of organo-nitrates from either α-pinene or β-pinene SOA hydrolyze at moderate relative humidities (~ 50% RH)."

3. Line 208-211: As the experimental conditions in the study by Takeuchi and Ng and the study by Claflin and Ziemann are different (e.g., RO2+NO3 dominant, low OA loading at ~60 ug/m3 and RO2+RO2 dominant, high OA loading on the order of hundreds of ug/m3, respectively), it is hard to directly use these studies to note that similar differences were observed between ESI and FIGAERO-CIMS in this work.

Additionally, because we are looking at different systems (α vs. β pinene) we decided to remove the sentence that mentions the difference between the ESI and FIGAERO from the different studies.

The removed sentence:

4. Line 352-353: As seed aerosols are not used in this work, perhaps the relative small amount of particle water could be the reason for negligible hydrolysis observed for dimer dinitrates? Also, it would be useful to note in Section 2.1 that seed aerosols are not used.

We have added a note about the possibility of the water content making a difference on Line 376:

"The lack of hydrolysis could come from the lack of water in the particles, which could differ from other experiments that have used seed aerosols (Takeuchi and Ng, 2019;Nah et al., 2016)."

We have added a note in the experimental section (Section 2.1) regarding the seed aerosol:

Line 79: "No seed particles were used in this study, in contrast to other studies that have been performed on similar systems (Takeuchi and Ng, 2019)."

References:

Claflin, M. S., and Ziemann, P. J.: Identification and Quantitation of Aerosol Products of the Reaction of β-Pinene with NO3 Radicals and Implications for Gas- and Particle-Phase Reaction Mechanisms, The Journal of Physical Chemistry A, 122, 3640-3652, 10.1021/acs.jpca.8b00692, 2018.

Nah, T., Sanchez, J., Boyd, C. M., and Ng, N. L.: Photochemical Aging of α-pinene and β-pinene Secondary Organic Aerosol formed from Nitrate Radical Oxidation, Environmental Science & Technology, 50, 222-231, 10.1021/acs.est.5b04594, 2016.

Takeuchi, M., and Ng, N. L.: Chemical composition and hydrolysis of organic nitrate aerosol formed from hydroxyl and nitrate radical oxidation of α-pinene and β-pinene, Atmos. Chem. Phys., 19, 12749-12766, 10.5194/acp-19-12749-2019, 2019.

---

## Author Response (AR2)

Dear Authors,

Thanks for addressing the reviewers and my comments. I just have one additional minor comment with respect to the question regarding Experiments #1 and #3 appeared to be conducted under identical conditions but with very different maximum SOA. It was noted that the short duration of N2O5 injection might have caused inhomogeneities in the chamber and possibly differences in the yields of formation. But it was also noted that since the mass spectra are similar, this suggested that the overall reaction pathway is similar. It is not clear how the yields of formation can be different but with similar overall reaction pathway? Perhaps while it was intended to have Experiments #1 and #3 conducted under identical conditions, for all practical purposes (and interpretation of the data, which show different results for these two experiments), one shall consider them as different experiments taking place under different conditions? I think it would be very helpful to include a short description in the experimental section to clarify these for the readers. Once this is addressed, the manuscript can be accepted for publication in ACP.

Best,
Sally

Dear Sally, (our response in blue, and the changed text is shown in orange)

Thank you for your careful reading and consideration of the experiments that we have conducted and presented in our manuscript.

We agree that the experimental conditions were intended to be identical, but as pointed out, the difference in SOA yields between the experiments means that they should not be considered to be exactly the same.

We have added the following lines to address your comment and to be consistent with this thread in the manuscript when discussing experiments 1 and 3 in the first results section:

Line 87: It was intended that Experiments 1 and 3 would be identical repeats since the α-pinene and $N_2O_5$ additions were effectively identical, however there was large differences in the mass loading observed. The reason for the difference in yield is not clear, and may result from inhomogeneities in the chamber during the short burst of $N_2O_5$. Therefore, even though experiments 1 and 3 were intended to be conducted under similar conditions, we cannot state unequivocally that they are identical.

And Line 227: Overall, the similarity in the measured composition in experiments 1 and 3 suggests these conditions were relatively similar, though they should not be considered exact replicates because of the difference in the SOA yield between the experiments (Table 1).